# Processing of AZ91D Magnesium Alloy by Laser Powder Bed Fusion

**Klára Nopová** [1], **Jan Jaroš** [2] , **Ondřej Červinek** [2] , **Libor Pantělejev** [1] , **Stefan Gneiger** [3] , **Sascha Senck** [4]
**and Daniel Koutný** [2,*]

1   Institute of Materials Science and Engineering, Brno University of Technology, Technická 2896/2,
    61669 Brno, Czech Republic
2   Institute of Machine and Industrial Design, Brno University of Technology, Technická 2896/2,
    61669 Brno, Czech Republic
3   AIT Austrian Institute of Technology GmbH, Light Metals Technologies Ranshofen,
    Lamprechtshausener Straße 61, 5282 Ranshofen, Austria
4   Research Group Computed Tomography, University of Applied Sciences Upper Austria, Stelzhamerstrasse 23,
    4600 Wels, Austria
*   Correspondence: daniel.koutny@vut.cz

**Abstract:** Magnesium alloys are perspective materials for use in transportation, aerospace and medical industries, mainly because of their good load-to-weight ratio, biocompatibility and biodegradability. For the effective production of magnesium components by the laser powder bed fusion (LPBF) process, the process parameters with verified mechanical properties need to be determined. In this paper, we prepared bulk samples with a high relative density of AZ91D magnesium alloy. Tensile tests were then performed on LPBF samples to evaluate the mechanical properties. Our results show that the bulk samples achieved a relative density >99%, in multiple planes over the full sample height, while the mechanical properties reached values of $Y_S$ = 181 MPa, *UTS* = 305 MPa and $A_{5.65}$ = 5.2%. The analysis by scanning electron microscope revealed fine β-$Mg_{17}Al_{12}$ particles in the microstructure, which have a positive effect on the mechanical properties. The chemical composition of magnesium alloy AZ91D changed slightly during processing by LPBF due to the evaporation of the Mg content. However, the resulting composition still corresponds to the range specified by the ASTM standard for the AZ91D alloy.

**Keywords:** AZ91D; magnesium alloy; laser powder bed fusion; energy dispersive spectroscopy; scanning electron microscope; process parameters; single tracks; porosity

## 1. Introduction

Nowadays, one of the trends in the transport industry is the reduction in the carbon footprint, targeting mainly vehicles and aviation. The most attractive way to reduce the carbon footprint is to reduce the weight of parts, which can be achieved effectively by combining lightweight materials with additive manufacturing technologies.

The advantages of combining lightweight materials with additive manufacturing technology are used to produce complex parts in the automotive and aerospace industries. One of the lightweight material groups consists of magnesium alloys, which have low weight, good energy absorption and thermal and acoustic properties [1]. In addition, magnesium alloys have good specific strength due to their light weight [2,3]. In particular, the magnesium alloy AZ31 is more durable than the aluminum alloy 2017A at low fatigue cycles with stress amplitudes of less than 150 MPa [3,4]. However, the processability of magnesium alloy is complicated due to the narrow temperature range between the melting and boiling points [5] and due to the high affinity of magnesium alloys for oxygen [6].

For the production of parts from magnesium alloys, casting is mainly used. Cast components have bad uniformity of the microstructure because the alloying elements

are segregated in the matrix of Mg, which is caused by the wide range of liquid phases between Mg and Al or Mg and Zn. Therefore, the produced part needs to be optimized by heat treatment. The disadvantages of this process are high cost and long cycle time, which are needed for the production of parts. These issues can be solved by using additive manufacturing technology, which allows the production of complex parts [7].

One of the additive technologies that is possible for the production of magnesium alloys is the LPBF process. LPBF uses a high-power laser to melt the metal powder into the shape of parts layer by layer [8,9]. Production of complex parts without imperfections requires an optimal combination of LPBF process parameters, i.e., laser power, laser speed, hatch distance and production strategy. In the case of magnesium alloys, the right combination of process parameters is more important because the high energy of laser power can cause the evaporation of magnesium, which leads to the formation of imperfections such as porosity [10–12].

Therefore, in this study, the LPBF process parameters are developed for the production of volume parts from magnesium alloy AZ91D. The DoE method is used to produce the cubic samples with different process parameter combinations. The best combination of process parameters is used for the production of tensile samples to obtain the mechanical properties of magnesium alloy AZ91D processed by LPBF technology.

## 2. Materials and Methods

Magnesium alloy AZ91D was processed by LPBF to obtain the optimal process parameters based on the high relative density. First, the basic experiment of single-track welds was performed to find out the optimal range of process parameters, i.e., laser power (LP) and laser speed (LS), for further production of bulk samples. The optimal range of process parameters was used for the design of volume process parameters by design of experiment (DoE) for the production of bulk cube samples. The porosity of samples was evaluated by metallographic analysis. The process parameters with the highest relative density were used for the production of bulk samples for tensile testing to determine the mechanical properties.

### 2.1. Fabrication of Samples

All samples were produced by an SLM 280$^{HL}$ machine (SLM Solutions Group AG, Lübeck, Germany) equipped with a 400 W ytterbium continuous wave fiber laser. The laser source has Gaussian power distribution focused on the spot size diameter of 82 μm. Argon inert gas was used to prevent the magnesium alloy AZ91D from oxidation. The fumes created by the vaporization of alloying elements of magnesium alloy AZ91D powder blur the laser and reduce the energy of the laser beam on the melting of the powder particles. Therefore, the more powerful gas pump (Becker, SV300/2, Cuyahoga Falls, OH, USA) was used to increase the suction of fumes. A layer thickness of 50 μm and a platform preheated to 135 °C were used in all experiments.

Magnesium powder AZ91D (Dome Metals Co., Ltd., Zhengzhou, China) was used for the experiments. The powder particles had a median diameter distribution of 38.6 μm, $D_{10}$ 21.8 μm and $D_{90}$ 56.9 μm (Figure 1). The chemical composition of powder particles determined by atomic absorption spectroscopy is described in Table 1 (values were supplied by powder producer).

**Table 1.** Chemical composition of magnesium alloy AZ91D powder.

| Element | Mg | Al | Zn | Mn | Si | Cu | Fe |
|---------|---------|------|------|------|------|-------|-------|
| Wt.% | Balance | 9.08 | 0.65 | 0.23 | 0.04 | <0.01 | <0.01 |

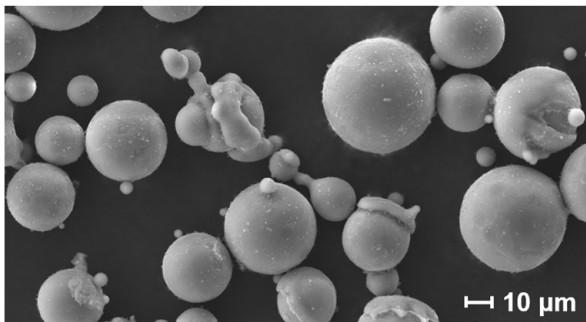

**Figure 1.** SEM image of magnesium alloy AZ91D powder.

## 2.2. Process Parameter Development

### 2.2.1. Single-Track Welds

Weld track shape is one of the key parameters for a reliable LPBF process. Due to the repeating nature of the layer-by-layer approach, the side-by-side overlapping and follow-up overmelting of weld tracks happen during the fabrication of the next tracks. In case the original single-track weld shows inconsistencies such as irregular shape and entrapped vapor cavities or instabilities such as interrupted track or even balling effect, the possibility of a reliable process resulting in a homogenous material with low porosity is very low.

The depth of the weld tracks (TD, Figure 2a) is given by the laser melting mode (conduction or keyhole) [13], which is dependent on a number of parameters such as the type of material, the size of the particles used, the volumetric energy density and the absorptivity of the material. The absorptivity and vaporization play an important role in the transition from conduction to keyhole mode [14]. During the keyhole mode, the weld track is typically more unstable, and incomplete collapse of the vapor cavity leaves voids in the wake of the laser beam [15]. The track height (TH, Figure 2a) plays an important role in the built-up perspective. The collapsed/deepened tracks or very-low-height tracks should be eliminated from further use because, in follow-up layers, new material would not be built up and only overmelting and the gradual evolution of defects would occur. The width of the track (TW, Figure 2a) drives the hatch distance parameter when the volumetric material is built. Typically, 30–50% overlapping is used between neighboring weld tracks.

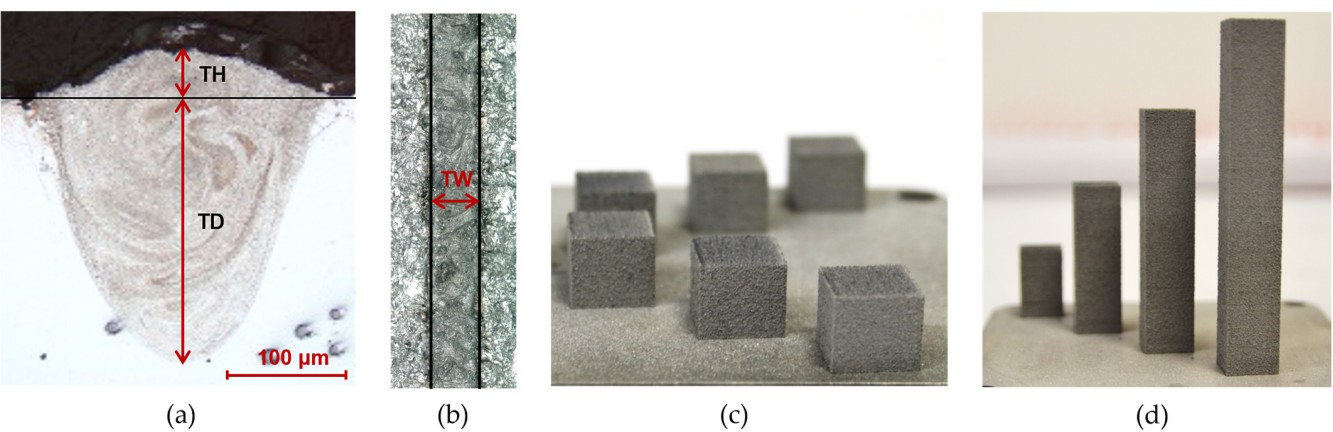

(a)  (b)  (c)  (d)

**Figure 2.** (**a**,**b**) Measured parameters of single track–track height (TH), track depth (TD), track width (TW), (**c**) bulk cube samples and (**d**) bulk cuboid samples.

The experiment with single-track welds was performed to determine the optimal combination of process parameters that leads to the production of weld tracks without imperfections. A wide range of process parameters was used, i.e., laser power of 150–210 W with a step of 10 W and laser speed of 325–750 mm/s with a step of 25 mm/s. Single tracks

were scanned to analyze the continuity and track widths (Figure 2b) by digital microscope (Keyence VHX-6000, Z250R lens, zoom 250×, Mechelen, Belgium). Then, metallographic analysis was performed to evaluate the geometry of the single tracks (Figure 2a), i.e., track height and track depth [16]. Metallographic analysis also provides insight into the quality of the produced tracks, revealing imperfections such as entrapped pores, cracks and large cavities, which should be excluded from further experiments.

### 2.2.2. Bulk Samples I

To evaluate the influence of process parameters on material quality, relative density is typically the main factor. Therefore, bulk cube samples with dimensions of 13 × 13 × 13 mm were produced (Figure 2c). The meander scanning strategy with rotation of laser track direction by 67° in each layer was used. The processing parameters to produce cube specimens were selected based on the design of experiment (DoE) methods. The boundary conditions were based on the optimal range of process parameters from the previous experiment, i.e., laser power in the range of 150–210 W, laser speed of 475–750 mm/s and overlap of 30–50%. The hatch distance is calculated as a percentage of overlap from the track width measured in the previous experiment. A total of 20 cube samples were produced in total.

### 2.2.3. Bulk Samples II

Two sets of process parameters were used to produce bulk cube samples with dimensions 20 × 20 × 20 mm to evaluate the formation of porosity in a larger volume. The first set of process parameters was selected based on the relative density of cube samples (Table 2, sample 4-1). The second set of process parameters is based on an analysis of process parameters and porosity values obtained from the cube experiment using DoE in the software MiniTab. Those process parameters for which the DoE predicts a porosity of only 0.04% are used (Table 2, sample 4-2). The porosity of samples was detected in three perpendicular planes, and the process parameters with higher relative density are used in the further experiment.

**Table 2.** Selected process parameters for the test of bulk samples.

| Bulk Sample | *LP* (W) | *LS* (mm/s) | *OL* (%) | *HD* (mm) | *RD* (%) |
|:---:|:---:|:---:|:---:|:---:|:---:|
| 4-1 | 180 | 612.5 | 40 | 0.133 | 99.4 ± 0.02 |
| 4-2 | 197 | 669 | 54 | 0.100 | ~100 |

Note: the last column in the table shows the expected values of porosity for bulk samples.

### 2.2.4. Bulk Samples III

The optimal process parameters based on the results from the previous experiment, i.e., laser power of 180 W, laser speed of 612.5 mm/s and overlap of 40%, were used to produce bulk cuboid samples for evaluation of relative density over the sample height. During sample production, many factors affect the quality of samples. In the case of magnesium alloy AZ91D, the production is mainly affected by fumes from melting the powder particles. Therefore, the experiment was designed to evaluate the stability of production by the SLM machine. Four samples with a base dimension of 12 × 12 mm and heights of 20, 40, 60 and 80 mm (Figure 2d) were produced. The stability of production was evaluated based on the relative density of samples with different heights.

### 2.3. Relative Density

The relative density was evaluated on the cross-sections of the cube and bulk samples. The samples were metallographically prepared by wet grinding and polishing and observed by a light microscope (Zeiss Axio Observer Z1m, Oberkochen, Germany). The relative density of cube samples was observed in the XY plane. The bulk samples were analyzed in the three perpendicular planes to obtain the porosity distribution in the sample. The bulk samples were divided into smaller parts and then analyzed in the XY and XZ planes

to obtain the pore distribution through the samples (Figure 3a). The relative density was measured by image analysis in the software ImageJ as a percentage of black color representing pores and white background color of the specimen material (Figure 3b,c).

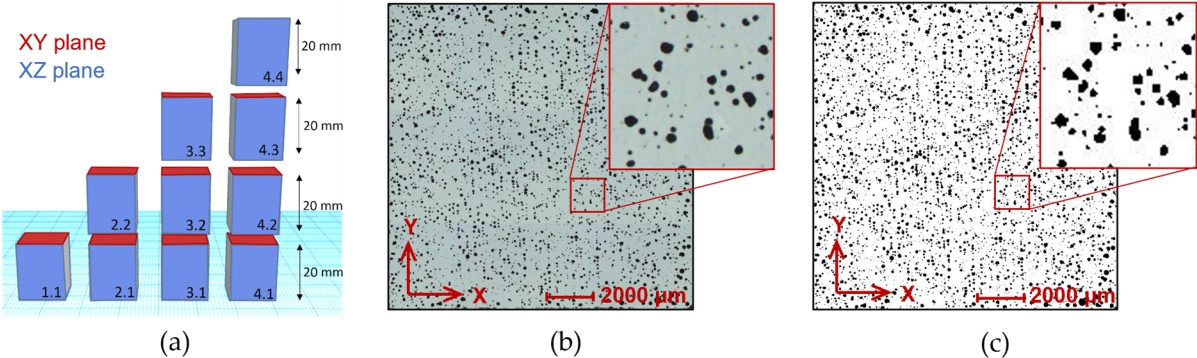

|     |     |     |
| :-: | :-: | :-: |
| (a) | (b) | (c) |

**Figure 3.** (**a**) Bulk sample divided into the sections for metallographic processing, (**b**) image of porosity in the sample after metallographic processing and (**c**) image from ImageJ software for evaluation of relative density.

### 2.4. Mechanical Properties

Samples with a dimension of 12 × 12 × 74 mm were produced in two separate jobs with optimal process parameters from the previous experiment (Bulk samples II). The samples were prepared by machining according to DIN 50125 (Form B), with a sample diameter of 6 mm and gage length of 30 mm. Machining was performed to obtain the required shape of the sample; furthermore, removing the surface layer of the sample reduces the influence of subsurface porosity. The two samples were tested in the as-build state by a universal testing machine (Zwick/Roell Z250, Zwick GmbH & Co. KG, Ulm, Germany) according to the EN ISO 6892-1 standard at room temperature.

### 2.5. Fractographical Analysis

After the tensile testing, the broken samples were used for fractographic analysis to determine the fracture mechanism of samples produced by LPBF. The analysis was performed using a scanning electron microscope (SEM–Zeiss Ultra Plus, Oberkochen, Germany).

### 2.6. Microstructural Analysis

The microstructure of tensile samples was analyzed, and the shape and distribution of pores were observed. Additionally, the solidification of the track pool was analyzed to describe the microstructure of produced material. The cross-sections of the broken tensile samples (out of the gauge length of the samples) were metallographically processed. The cross-sections samples were etched by Nital 1% compound and evaluated using a light microscope (Olympus GX51, Olympus, Tokyo, Japan).

### 2.7. Chemical Composition

Due to the higher susceptibility to evaporation of alloying elements during the LPBF process, the chemical composition of powder and LPBF processed material was analyzed by energy dispersive spectroscopy (EDS).

### 3. Results and Discussion

#### 3.1. Single-Track Welds

Based on the literature review described in Section 2.2, a map of the process parameters for the experiment is drawn up (Figure 4a). The map is divided into three areas depending

on the energy range achieved for each single-track group. The energy value *E* is determined by Equation (1).

$$E = \frac{LP}{LS \cdot LT}, (J/mm) \tag{1}$$

where *LP* is the laser power (W), *LS* is the scanning speed (mm/s) and *LT* is the layer thickness (mm).

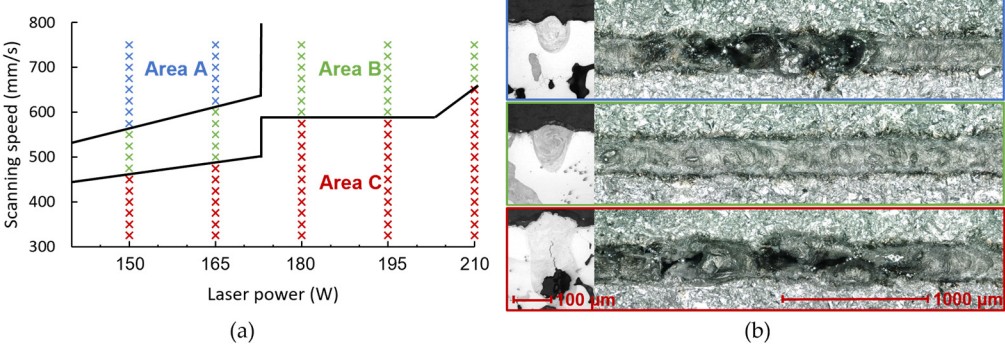

(a)                                                                                    (b)

**Figure 4.** (**a**) Map of process parameters for single-track test with defined areas and (**b**) representative single track for a specific area (metallographic image and 3D digital microscope image).

The individual single-track welds were inspected to identify the parameters that resulted in high-quality single tracks. Such parameters were considered to be suitable for further investigation. They are characterized by the optimal shape of the cross-section and the continuity of the track. The continuity of the track is affected by significant contractions observed in the entire volume of the single-track series. Based on the inspection, the main characteristics of the single tracks in each area are identified as follows:

- A—The region with a low value of input energy *E* = 3.7–4.7 J/mm shows a suitable depth of 179 ± 25 μm to connection with previous layers and a width of 204 ± 11 μm of the single track; however, the tracks in this region are not continuous. There are often areas where the material is not completely melted (unmelted powder particles occurred, Figure 4b—blue frame), which significantly affects both the continuity and the height of the track (32 ± 14 μm).

- B—The region with an average input energy *E* = 4.7–6.5 J/mm shows no cracks or pores around the weld (Figure 4b—green frame). The track depth is 218 ± 21 μm, the track height is 22 ± 2 μm and the track width is 215 ± 6 μm. Due to the relatively uniform dimensions of the track and the continuity of all tracks in this area, area B is evaluated as suitable for further testing.

- C—The region with a high value of the input energy value *E* = 6.5–12.7 J/mm shows an excessive depth of 338 ± 81 μm and a width of 233 ± 20 μm of single tracks. Metallographic analysis reveals the frequent occurrence of cracks and pores both in the deposit and in the surrounding material (Figure 4b—red frame), while the continuity of the deposit is not guaranteed.

Based on the analyses and their comparison, 27 process parameters were evaluated as suitable for further testing (Figure 5). These are single tracks in the green area that show good cross-section shape and continuity. The balling effect, pores and cracking are not observed in them. These factors indicate a stable behavior of the weld tracks in their entire sample length. Similar results were obtained in the studies [17–19] for the magnesium alloys, where three areas that characterize almost identical defects in individual welds are also observed.

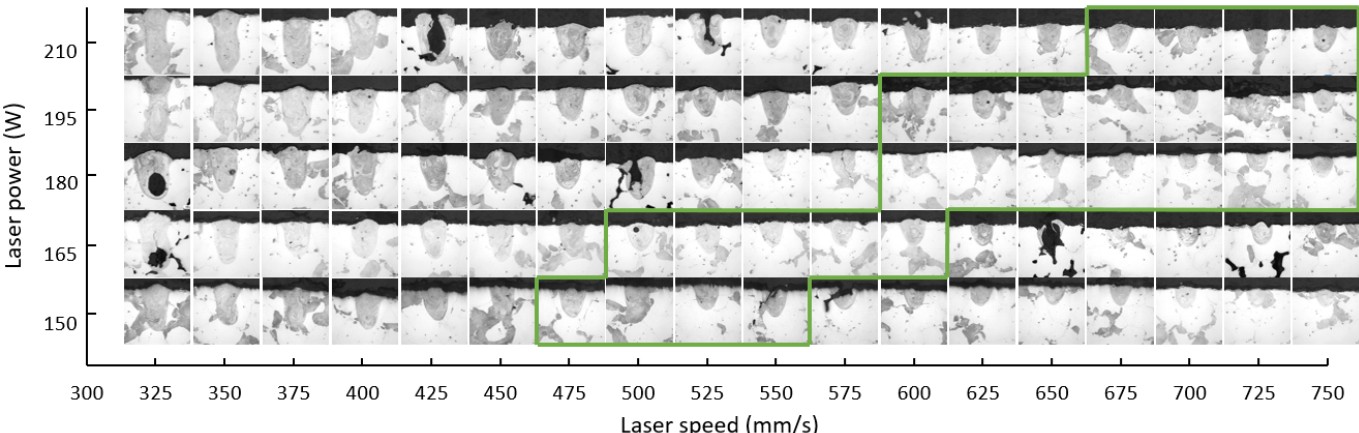

**Figure 5.** Metallographic images of single tracks for the entire experimental area. Green border represents parameters used for follow-up testing.

### 3.2. Bulk Samples I

In the following step, cube samples were produced in three series to minimize the negative effects of the sample position of the platform. Using DoE, a total of 20 process parameters were used, which are listed in Table 3. *LP* and *LS* from the previous single-track test were chosen as the DoE input marginal parameters (Section 2.1). Furthermore, the distance between the individual tracks ('Hatch distance'—*HD*; Figure 6) and the percentage of overlap of individual tracks ('Overlap'—*OL*) are included in DoE. *HD* is calculated using *OL* and the width of the tracks from the single-track weld test. In addition, the table shows the values of the input energy density $E_v$, which is determined according to Equation (2), and the resulting values of relative areal density (*RD*) of the material:

$$E_v = \frac{LP}{LS \cdot LT \cdot HD} \left[ J / mm^3 \right] \qquad (2)$$

**Table 3.** Overview of process parameters and measured values of area relative density (*RD*) of cube samples with standard deviation, XY plane.

| Group | Sample | *LP* (W) | *LS* (mm/s) | *OL* (%) | *HD* (mm) | $E_v$ (J/mm³) | *RD* (%) |
|---|---|---|---|---|---|---|---|
| | 1-1 | 150 | 750 | 50 | 0.098 | 40.8 | 99.2 ± 0.02 |
| | 1-2 | 150 | 475 | 30 | 0.151 | 41.8 | 99.0 ± 0.02 |
| | 1-3 | 180 | 612.5 | 40 | 0.133 | 44.2 | 99.2 ± 0.12 |
| 1 | 1-4 | 210 | 475 | 50 | 0.126 | 70.2 | 98.0 ± 0.02 |
| | 1-5 | 180 | 612.5 | 40 | 0.133 | 44.2 | 98.9 ± 0.11 |
| | 1-6 | 210 | 750 | 30 | 0.153 | 36.6 | 98.8 ± 0.01 |
| | 1-7 | 180 | 612.5 | 40 | 0.133 | 44.2 | 99.1 ± 0.01 |
| | 2-8 | 150 | 475 | 50 | 0.108 | 58.5 | 98.4 ± 0.03 |
| | **2-9** | **210** | **475** | **30** | **0.176** | **50.2** | **92.8 ± 0.05** |
| | 2-10 | 210 | 750 | 50 | 0.110 | 50.9 | 98.4 ± 0.02 |
| 2 | 2-11 | 150 | 750 | 30 | 0.137 | 29.2 | 94.8 ± 0.11 |
| | 2-12 | 180 | 612.5 | 40 | 0.133 | 44.2 | 98.8 ± 0.04 |
| | 2-13 | 180 | 805 | 40 | 0.120 | 37.3 | 97.2 ± 0.06 |
| | 2-14 | 138 | 612.5 | 40 | 0.120 | 37.6 | 97.6 ± 0.08 |
| | 3-15 | 180 | 612.5 | 26 | 0.164 | 35.8 | 99.0 ± 0.01 |
| | 3-16 | 222 | 612.5 | 40 | 0.135 | 53.7 | 99.0 ± 0.01 |
| | 3-17 | 180 | 612.5 | 54 | 0.102 | 57.6 | 99.3 ± 0.21 |
| 3 | 3-18 | 180 | 420 | 40 | 0.144 | 59.5 | 97.7 ± 0.04 |
| | **3-19** | **180** | **612.5** | **40** | **0.133** | **44.2** | **99.4 ± 0.02** |
| | 3-20 | 180 | 612.5 | 40 | 0.133 | 44.2 | 99.0 ± 0.02 |

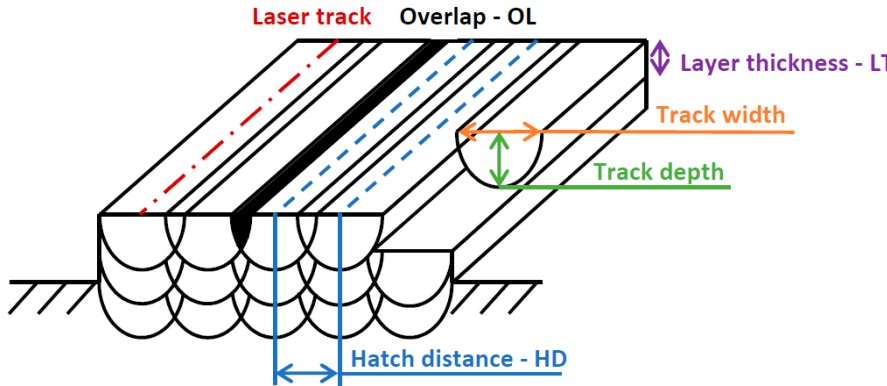

**Figure 6.** Parameters of tracks in cube sample [20].

Representative images of the samples with the lowest and highest relative densities are shown in Figure 7. The highest relative density was achieved in sample 3-19 with the process parameters $LP$ = 180 W, $LS$ = 612.5 mm/s, $HD$ = 0.133 mm and $OL$ = 40%. These process parameters are used several times to avoid calculation and measurement errors in production of only one sample. On the other hand, sample 2-9 shows extensive porosity (Figure 7a) mainly consisting of spherical pores and voids with a size of 10–200 μm. The pores and voids are irregularly distributed throughout the sample volume. Local line alignment of the cavities can be observed, indicating the correlation of their formation with the laser beam track as a consequence of high input energy density [21,22]. The porosity of sample 3-19 is minimal (Figure 7b). Spherical pores with a size of about 10 μm occur mainly in the subsurface region of the sample. The AZ91D alloy investigated in the study by Wei et al. [23] shows a similar result with a relative density of 99.52% and an input energy density $E_v$ = 166.7 J/mm$^3$. In this case, the $E_v$ value is several times higher than in the case of sample 3-19 ($E_v$ = 44.19 J/mm$^3$). The clear difference between the $E_v$ values and the almost identical relative density can be explained by several factors. First of all, the different pieces of equipment used to produce samples must be taken into account. Additionally, the method of calculating $E_v$ should be taken into consideration (Equation (2)), as the same $E_v$ values can be obtained with different input parameters. Therefore, it is always necessary to compare the process parameters themselves. For sample 3-19, these are: $LP$ = 180 W, $LS$ = 612.5 mm/s, $HD$ = 0.133 mm and $LT$ = 0.05 mm, while in the case of Wei et al. [23], they are $LP$ = 200 W, $LS$ = 333.3 mm/s, $HD$ = 0.09 mm and $LT$ = 0.04 mm. From the process parameters, it can be seen that despite an almost identical relative density of the material, different microstructural properties can be expected.

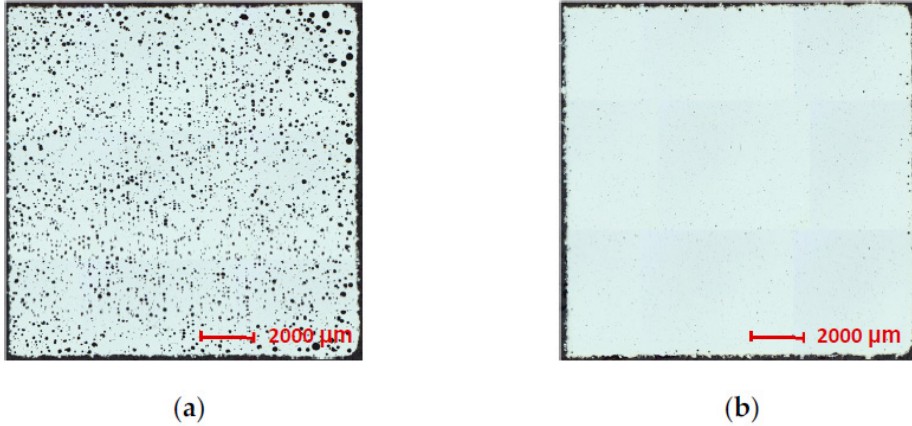

(a)  (b)

**Figure 7.** Cross-section images of the samples (**a**) 2-9 (*RD* = 92.8%); (**b**) 3-19 (*RD* = 99.4%).

### 3.3. Bulk Samples II

The following test was intended to show the influence of the size (volume) of the sample on the occurrence of defects in the material when the process parameters from the previous experiment are used. Therefore, based on the test with cubic samples, two sets of process parameters were selected and tested on bulk samples. The first set was chosen as the parameter set that leads to the minimum porosity values for the cube tests (Table 2, sample 4-1). The second parameter set was proposed based on the DoE as parameters for estimating the minimum resulting porosity (Table 2, sample 4-2).

After preparation, the porosity of the samples was evaluated in three perpendicular planes according to Section 2.3. Table 4 shows the measured values of porosity for the individual planes of both samples. Contrary to expectation, both samples contain a considerable number of pores and cavities distributed throughout their volume. In contrast, no cracks are observed in any of the samples tested. Due to the lower average porosity and the lower generation of fume gas during the SLM process, the parameters used for the production of sample 4-1 were chosen as input for the following test.

**Table 4.** Porosity measured in different planes and relative density.

| Bulk Sample | Porosity (%) | | | | RD (%) |
|---|---|---|---|---|---|
| | Plane XY | Plane YZ | Plane XZ | Average | |
| 4-1 | $2.1 \pm 0.12$ | $5.2 \pm 0.34$ | $3.7 \pm 0.22$ | 3.7 | $96.3 \pm 1.26$ |
| 4-2 | $4.1 \pm 0.25$ | $3.3 \pm 0.15$ | $4.7 \pm 0.39$ | 4.0 | $96.0 \pm 0.57$ |

### 3.4. Bulk Samples III

The following series of samples were used to detect porosity changes at different heights in bulk samples. For the production, the parameter set from Table 2 sample 4-1 was used because, in the previous experiment, it led to lower porosity. After production, the individual samples were cut and metallographically prepared for the observation of porosity with a light microscope. Image analysis was used to evaluate the relative surface density of the samples in the XY planes at different sample heights (Figure 2d). An overview of the measured values can be found in Table 5. The marking of the samples corresponds to Figure 3a. Larger spherical pores at all height levels are observed in the subsurface area of the samples. The regular arrangement of pores in the subsurface area is usually the result of a local increase in energy at the end of the individual hatching vectors, which is caused by the deceleration at the end of the vector and the acceleration at the beginning of the neighboring vector (turnaround point), which can be removed with the use of so-called skywriting in newer scanning units [24]. Only very small gas pores of about 10 μm size are observed in the volume of the sample, as shown in Figure 8. A specific or significantly different pore arrangement in the individual height levels was not observed.

**Table 5.** The resulting values of the porosity of the XY planes.

| Height | | | | |
|---|---|---|---|---|
| 80 mm | | | | **5-4.4** 0.7% |
| 60 mm | | | **5-3.3** 0.5% | **5-4.3** 0.5% |
| 40 mm | | **5-2.2** 0.5% | **5-3.2** 0.4% | **5-4.2** 0.9% |
| 20 mm | **5-1.1** 1% | **5-2.1** 0.7% | **5-3.1** 0.5% | **5-4.1** 1.1% |
| **Average porosity** | 1% | 0.6% | 0.5% | 0.8% |

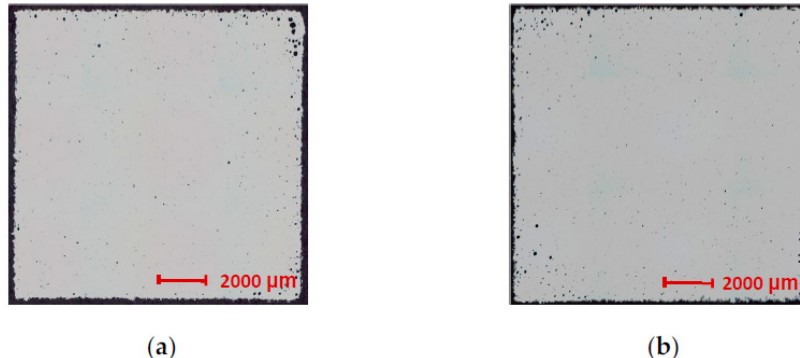

**(a)**          **(b)**

**Figure 8.** Test of bulk samples II, plane XY, (**a**) 5-4.1; (**b**) 5-4.4.

Table 6 shows the values of the relative density of the XZ planes in the individual samples (Figure 3a). The lower part of the samples (0–20 mm above the platform) shows a higher percentage of porosity than the area of the samples in the height plane > 20 mm above the platform. Figure 9 compares the lower (Figure 9a) and upper area of sample 5-4 (Figure 9b). In the lower area, a more frequent occurrence of larger spherical pores (100 ± 20 µm) is observed, mainly in the subsurface area of the sample, in contrast to the upper part of the sample where only the occurrence of small pores (10 ± 6 µm) is observed. Total porosity is also significantly higher in the lower areas. On the other hand, no significant influence of the sample size on the resulting porosity is observed, in contrast to the study published by Phutela et al. [25], where a significant decrease in porosity is detected with increasing sample size for the alloy Ti–6Al–4V.

**Table 6.** The resulting values of the porosity of the XZ planes.

| **Height Range** | | | | |
| --- | --- | --- | --- | --- |
| 60–80 mm | | | | **5-4.4** 0.5% |
| 40–60 mm | | | **5-3.3** 0.7% | **5-4.3** 0.4% |
| 20–40 mm | | **5-2.2** 0.7% | **5-3.2** 0.5% | **5-4.2** 0.6% |
| 0–20 mm | **5-1.1** 1% | **5-2.1** 1.4% | **5-3.1** 0.7% | **5-4.1** 1.7% |
| **Average porosity** | 1% | 1% | 0.6% | 0.8% |

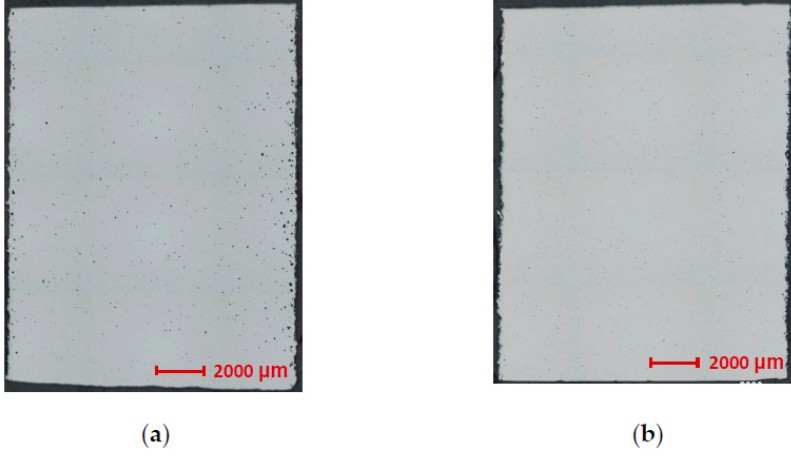

**(a)**          **(b)**

**Figure 9.** Test of bulk samples II, plane XZ, (**a**) 5-4.1; (**b**) 5-4.4.

### 3.5. Tensile Testing

To assess the mechanical properties, the tensile tests were carried out at room temperature. For the fabrication of the samples, the process parameters from the previous test are retained. Table 7 shows the mechanical properties of the AZ91D alloy obtained by experiments and from the literature (Ys—0.2% proof stress, UTS—ultimate tensile strength and $A_{5.65}$—% elongation at break). Sample 7-1 has a higher ductility value compared to the other samples. The low mechanical properties of sample 6-1 can be attributed to defects that occurred during the SLM process. Wei et al. [23] attribute the different tensile test results of the studied alloy AZ91D to the number of pores present in the microstructure and the occurrence of the β-$Mg_{17}Al_{12}$ phase, which increases with increasing $E_v$ value. The different properties of sample 7-1 can be attributed to the specific value of $E_v$ and related amount of β-$Mg_{17}Al_{12}$ phase as was discussed by Wei [23] and Niu [21]. The $E_v$ value of sample 7-1 is 44.2 J/mm$^3$, the sample prepared by Wei [23] (SLM) is 166.7 J/mm$^3$ and the sample prepared by Niu [21] (SLM) is 277.8 J/mm$^3$. Therefore, a different amount of β-$Mg_{17}Al_{12}$ in the material is expected, influencing the resulting mechanical properties, as reported in the study by Wei [23].

**Table 7.** Results of tensile test performed at room temperature for AZ91D.

|  | Ys (MPa) | UTS (MPa) | $A_{5.65}$ (%) |
|---|---|---|---|
| **6-1** | 182 | 196 | 0.3 |
| **7-1** | 181 | 305 | 5.2 |
| **Wei SLM** [23] | 254 | 296 | 1.83 |
| **Niu SLM** [21] | 225 ± 5 | 306.5 ± 8.4 | 2 ± 0.5 |
| **Wei Cast** [23] | 160 | 225 | 3 |

### 3.6. Fractographic Analysis

Fractographic analysis was performed on the broken samples after tensile testing. Numerous unmelted powder particles and pores were visible on the fracture surface of sample 6-1 (Figure 10a). This phenomenon can be attributed to a local defect that occurred during the fabrication of sample 6-1 when there was a short job interruption caused by a pressure drop in the chamber and the fumes were not properly extracted before the subsequent laser processing began. The accumulated fumes caused the laser beam to scatter and resulted in the material not being sufficiently melted. Outside the defect area, the damage mechanism is of low-energy ductile character with a fine dimple morphology (Figure 10b).

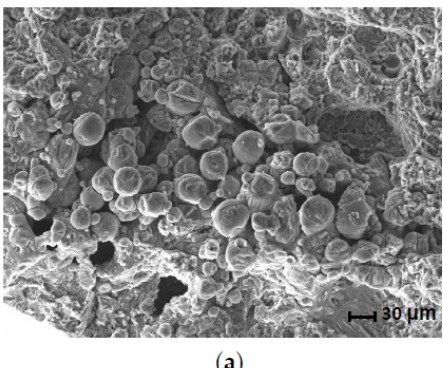
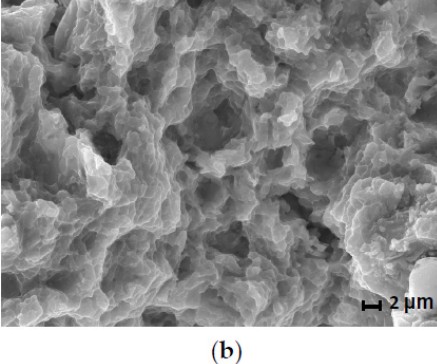

(**a**)　　　　　　　　　　　　　　　　　　　　(**b**)

**Figure 10.** Fracture surface of sample 6-1: (**a**) unmelted powder particles and pores; (**b**) fine dimple morphology, powder particle.

Local pores (Figure 11a) and hints of cleavage facets (Figure 11b) were observed on the fracture surface of sample 7-1, which exhibited fine dimple morphology. The morphological

features of the fracture surface indicate a low-energy ductile fracture. Similar results are obtained by Liu et al. [26] for the alloy AZ61.

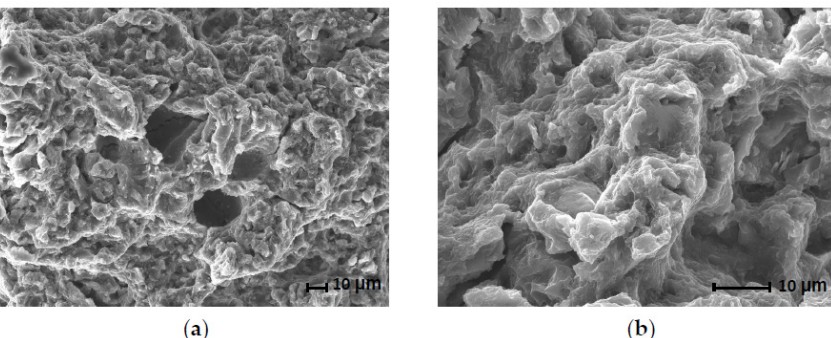

**(a)**       **(b)**

**Figure 11.** The fracture surface of sample 7-1: (**a**) spherical pores with cracks; (**b**) fine dimple morphology with hints of cleavage facets.

### 3.7. Microstructural Analysis

Two metallographic sections were made of the thread heads of sample 7-1. Initially, the first section was performed in the longitudinal direction (*L*) and the second in the transverse direction (*T*) in relation to the central (macro) axis of the sample. In the following step, the relative areal density was measured in these samples. The longitudinal sample has a porosity of 0.6% and the transverse sample of 0.3%. In both cases, the pores were mostly small and of irregular shape.

The microstructure can be seen in both sections in the etched state in Figure 12. Figure 12a shows solidified melt pools. Individual pools overlap each other, indicating good metallurgical bonding between adjacent layers. In the cross-section (Figure 12b), individual laser trajectories are visible. The trajectories are not continuous, which is due to a change in orientation during the scanning of the individual layers. Similar results are obtained in other studies [1,23,26]. The irregular shape of the pores in the material is visible in the metallographic images. The cavities are predominantly very small (approximately 10 μm) and distributed over the entire volume of the sample.

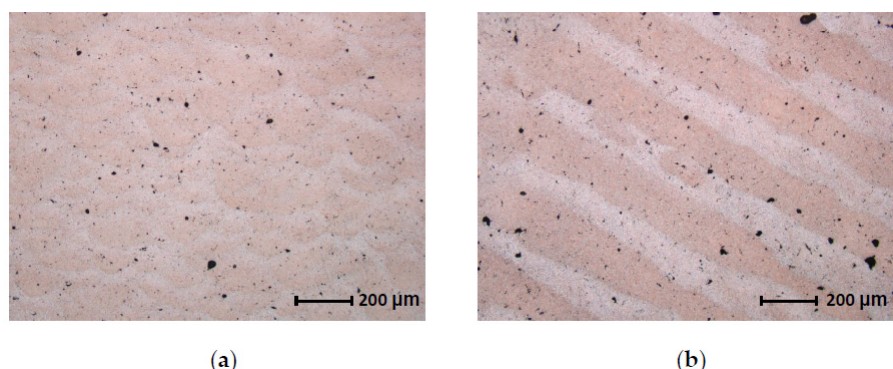

**(a)**       **(b)**

**Figure 12.** Sample 7-1: (**a**) longitudinal-L section; (**b**) transverse-T section.

For a detailed assessment of the microstructure, an analysis using a scanning electron microscope (SEM) was carried out. Figure 13 shows the very fine microstructure of the material, consisting of the α-Mg matrix, divorced β-$Mg_{17}Al_{12}$ phase at the grain boundaries and very small, irregularly arranged, Mn-rich precipitates [27,28]. The image shows the difference in grain size in the area of the center of the solidified melt pool and at its edge (fusion boundary), where the microstructure is much finer. To determine the actual chemical composition of individual phases, energy dispersive spectroscopy (EDS) was performed (Table 8).

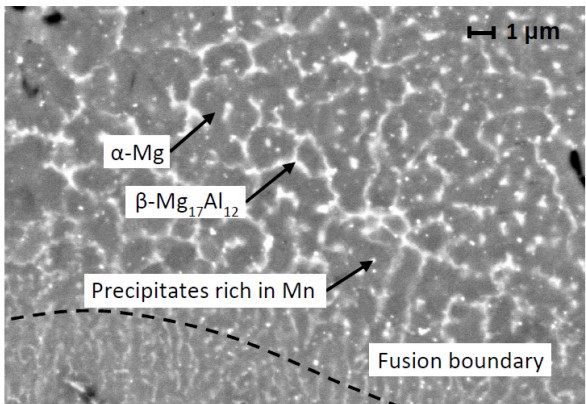

**Figure 13.** SEM image of microstructure in longitudinal section.

**Table 8.** Results of EDS chemical microanalysis of individual phases [wt.%].

|  | Mg | Al | Zn | Mn |
|---|---|---|---|---|
| **Matrix $\alpha$-Mg** | 94.33 | 5.35 | 0.32 | - |
| **Eutectic-phase β-Mg$_{17}$Al$_{12}$** | 80.27 | 15.78 | 1.40 | - |
| **Precipitates rich in Mn** | 79.35 | 15.31 | 1.23 | 4.11 |

### 3.8. Chemical Composition

For the LPBF processed material, the chemical composition analysis of a sample area of ~14,000 $\mu m^2$ was performed using EDS. Table 9 shows a decrease in Mg content, which is attributed to the evaporation of this element during the SLM process [29]. The magnesium has a low boiling point (1107 °C) and low evaporation heat (5.272 kJ/kg) at ambient pressure [5,30]. Evaporation of Mg causes a shift of other constituents to higher values [1,23]; thus, the overall composition slightly changes, but is still in the range of the ASTM standard for AZ91D alloy. The accuracy of the EDS measurement method must also be considered when assessing constituents' change (detection limit of the EDS method is approx. 0.1 wt.%) [31].

**Table 9.** Results of EDS chemical analysis of the material before and after the SLM process [wt.%].

|  | Mg | Al | Zn | Mn |
|---|---|---|---|---|
| **EDS—powder** | 90.46 | 8.58 | 0.69 | 0.27 |
| **EDS—after SLM** | 90.22 | 8.64 | 0.89 | 0.25 |
| **ASTM B94-18 standard** | Bal. | 8.3–9.7 | 0.34–1.0 | 0.15–0.5 |

### 3.9. Future Outlook

As the relative density results of *bulk samples III* show a fairly even distribution of porosity over height, the mechanical properties exceed those achievable by the casting process, and the values are still comparable to recent studies, future research will be aimed at assessing the fatigue properties. Research has shown that some lightweight materials fabricated by the LPBF process can have good high-cycle fatigue properties [32]. The S-N curves show that they can withstand over $5 \times 10^7$ cycles without sample failure [33]. With optimization of the main process parameters such as laser power, laser speed, hatch distance and laser defocusing, control of the fatigue life can be reached [34]. Furthermore, the results of *bulk samples II* suggest that the energy density should be optimized for each part geometry (thin, thick and overhanging) to achieve uniform microstructure and properties [35].

## 4. Conclusions

The research performed in this study revealed the basic possibilities of AZ91D alloy processing using the LPBF process. During the research, process parameters were investigated, leading to the most suitable configuration. The resulting parameters led to a fine internal structure for the volume sample, competitive mechanical properties and minimization of internal porosity. Based on the abovementioned experiments and analyses, the following findings have been concluded:

- The process parameters of the SLM technology for the alloy AZ91D were developed and tested. With the developed parameters, a relative density of 99% and more was achieved for the cube samples. The final configuration, which resulted in low porosity and a fine, uniform microstructure, consisted of the following parameters: $LP$ = 180 W, $LS$ = 612.5 mm/s, $HD$ = 0.133 mm and $LT$ = 0.05 mm.
- The results of the tensile test showed very good mechanical properties of the material ($Y_s$ = 181 MPa, $UTS$ = 305 MPa and $A_{5.65}$ = 5.2%). Compared to the literature data, sample 7-1 showed more than twice higher ductility.
- Fractographic analysis performed after the tensile tests revealed that the damage mechanism is of low-energy ductile character with a fine dimple morphology.
- Light microscopy of the etched samples was used to obtain basic information about the microstructure of the processed material. The microstructure showed the typical structure of solidified melt pools in the longitudinal direction. In addition, the directivity of the solidified pools of the given laser trajectory was observed in the transverse direction.
- A detailed SEM analysis revealed a fine $\beta$-$Mg_{17}Al_{12}$ phase presence within the solidified melt pools and in the areas of the fusion boundaries separating the individual pools. EDS microanalysis was used for the determination of the chemical composition of the phases in the processed material.
- The studied alloy AZ91D showed a loss of wt.% of Mg after LPBF processing due to evaporation. The other alloying elements did not significantly change, and the resulting chemical composition was in the range of the ASTM standard for AZ91D alloy.

**Author Contributions:** Conceptualization, J.J. and O.Č.; methodology, K.N.; software, K.N.; validation, O.Č., J.J., L.P. and D.K.; formal analysis, K.N.; investigation, K.N.; resources, S.G. and S.S.; data curation, J.J.; writing—original draft preparation, K.N., O.Č. and J.J.; writing—review and editing, K.N., J.J., O.Č., D.K., L.P., S.G. and S.S.; visualization, J.J.; supervision, L.P. and D.K.; project administration, L.P. and D.K.; funding acquisition, D.K., S.G. and S.S. All authors have read and agreed to the published version of the manuscript.

**Funding:** This research was funded by the European Commission within the framework of the INTERREG V-A Austria–Czech Republic in the project "ReMaP" (Interreg project no. ATCZ229).

**Institutional Review Board Statement:** Not applicable.

**Informed Consent Statement:** Not applicable.

**Data Availability Statement:** Not applicable.

**Conflicts of Interest:** The authors declare no conflict of interest.

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
