# Peer review of "Processing of AZ91D Magnesium Alloy by Laser Powder Bed Fusion"

_applsci, doi:10.3390/app13031377_

Round 1
Reviewer 1 Report
In the paper, the authors have researched the process parameters of Laser Power Bed Fusion for AZ91D Magnesium Alloy. The bulk samples reach a relative density of > 99 %, which contribute to the good mechanical properties. However, the authors should address the following comments before acceptance.
1. The scale should be given in the figure in section 2.1 and the figure should be named Figure 1.
2. What is “laser powder”? Is laser power?
3. More information about the laser source used is needed. For example, the pulse duration of the laser and the spot size used are not mentioned.
4. There is a grammatical error in the sentence “To assess the mechanical properties, tensile test are carried out at room temperature.”.
5. The decrease in wt. % of Zn and Mg was caused by elements evaporation. However, the melting points of Al and Mg are very similar. Why does the wt. % of Al increase?
6. The conclusion of “The Mn content dropped even more from 0.89 % in the powder to only 0.27 % in the processed material” contradicts the date in Table 9.
7. The authors concluded that the ductility of sample 7 in this paper was twice as high as that of the samples in the literature. However, the strong proof of ductility performance is elongation and not ultimate tensile strength. Thus the elongation should be given.
8 Some import literature regarding LPBF of Mg should be referenced, like: Microstructure development and biodegradation behavior of additively manufactured Mg-Zn-Gd alloy with LPSO structure, Journal of Materials Science & Technology, 2023, 144.
Author Response
Dear Reviewer, The authors would like to thank you for your valuable comments which have helped to improve the quality of the paper. We have gone through all your comments and have incorporated them into the revised text of the paper. Please find below our response to each comment.
- The scale should be given in the figure in section 2.1 and the figure should be named Figure 1.
The figure of powder particles was removed from the Table 1, named as Figure 1 and proper scale was added.
- What is “laser powder”? Is laser power?
This was a typo, which unfortunately none of the authors discovered due to blind assumption while reading their own text. The “Laser powder” term was corrected to laser power throughout the article. The only term, “Laser powder“ remain in the name of the technology (Laser Powder Bed Fusion).
- More information about the laser source used is needed. For example, the pulse duration of the laser and the spot size used are not mentioned.
Missing information about the laser were added in section 2.1. The continuous wave (CW) fiber laser source with Gaussian power distribution focused on the spot size diameter of 82 μm was used in the research.
- There is a grammatical error in the sentence “To assess the mechanical properties, tensile test are carried out at room temperature.”.
This grammatical error was corrected in section 3.5. Several other sentences were corrected throughout the article regarding grammar and clarity of meaning. All corrections are marked in yellow.
- The decrease in wt. % of Zn and Mg was caused by elements evaporation. However, the melting points of Al and Mg are very similar. Why does the wt. % of Al increase?
We agree with the reviewer that the melting temperatures of pure Mg and Al are very close (649 °C vs 660.1 °C), however, it should be mentioned that to assess the evaporation of individual metals, it is also necessary to take into account the boiling temperature, which in case of Mg is 1090 °C, while for Al it is a much higher temperature of 2520 °C [R1]. The narrow temperature range between melting and boiling point is the reason for the more intensive evaporation of Mg.
Moreover, intensive evaporation is amplified by another material property – vapor pressure, which is naturally different for Mg and Al [R2, R3]. Considering the range of melting temperatures of the AZ91D alloy [R4] used in this study, then for the melting temperature ~ 500 °C we can calculate the vapor pressure using standard function [R1] according to the modification mentioned in [R5], the calculated vapor pressure of Mg is 1.18×10- 4 atm, whereas for Al it is only 1.77×10-15 atm. It supports the above-mentioned statement that Mg is more susceptible to evaporation than Al at the same temperature.
Regarding the measured data listed in Table 9, however, the accuracy of the used measurement method must also be considered when assessing element loss. The detection limit of the EDS method is approx. 0.1 wt. %, while the difference in the Al content of the powder and the SLM sample is only 0.06 wt. %, thus below the resolution of the method, and the Al content must therefore be considered "the same" within the measurement scatter/detection limit [R6].
[R1] GALE, William F, Terry C TOTEMEIER a T. C TOTEMEIER. Smithells Metals Reference Book. 8th ed. /edited by W.F. Gale, T.C. Totemeier. Jordan Hill: Elsevier Science & Technology, 2004. ISBN 0750675098
[R2] Wan-neng Zhang, Lin-zhi Wang, Zhong-xue Feng, Yu-ming Chen, Research progress on selective laser melting (SLM) of magnesium alloys: A review, Optik, Volume 207, 2020, 163842, ISSN 0030-4026, https://doi.org/10.1016/j.ijleo.2019.163842
[R3] Andrzej Pawlak, Maria Rosienkiewicz, Edward Chlebus, Design of experiments approach in AZ31 powder selective laser melting process optimization, Archives of Civil and Mechanical Engineering, Volume 17, Issue 1, 2017, Pages 9-18, ISSN 1644-9665, https://doi.org/10.1016/j.acme.2016.07.007
[R4] Standard Specification for Magnesium-Alloy Die Casting, ASTM International, B97-18, pp.1-6
[R5] Zhao, H., Debroy, T. Weld metal composition change during conduction mode laser welding of aluminum alloy 5182. Metall Mater Trans B 32, 163–172 (2001). https://doi.org/10.1007/s11663-001-0018-6
[R6] GOLDSTEIN, I. Joseph. Scanning electron microscopy and X-ray microanalysis. 3rd ed. New York, 2003, p. 355. ISBN 0-306-47292-3.
- The conclusion of “The Mn content dropped even more from 0.89 % in the powder to only 0.27 % in the processed material” contradicts the date in Table 9.
We would like to thank you again because during checking the chemical composition in the original measured data we found a mistake. The analyzed wt. % values of Zn and Mn elements were accidentally mixed up during rewriting the results to table 9. The correct values of Zn and Mn are 0.89 wt. % (Zn) and 0.25 wt. % (Mn). After correction of the values in section 3.8 and Table 9, we also changed the statement in the conclusion, because the overall chemical composition is still in the range of the ASTM standard for AZ91D alloy.
- The authors concluded that the ductility of sample 7 in this paper was twice as high as that of the samples in the literature. However, the strong proof of ductility performance is elongation and not ultimate tensile strength. Thus the elongation should be given.
We have to defend our statements, because we are describing the ductility by % elongation at break, signed as A5.56 according to international standard EN ISO 6892-1. Regarding this approach, we stated that sample 7-1 exhibited % elongation at break of 5.2 %, and in [15] the ductility value was 2+/-0.5 %, so that is why we stated that in the case of sample 7-1 the „ductility value“ is more than twice higher. For better clarity, we improved the appropriate text and added the missing sign „%“ into the text of paragraph 3.5. Tensile testing, i.e. (…..A5.56 – % elongation at break).
- Some important literature regarding LPBF of Mg should be referenced, like: Microstructure development and biodegradation behavior of additively manufactured Mg-Zn-Gd alloy with LPSO structure, Journal of Materials Science & Technology, 2023, 144.
The recommended paper was included in section 3.7. The identical microstructure is achieved in the recommended article as in the case of this study.
Reviewer 2 Report
The manuscript entitled “applsci-2150217-LB-PBF” dealing with AM has been reviewed. The paper has been nicely written but needs significant improvement. Please follow my comments.
1. How authors provided the information in Table 1. Chemical composition of magnesium alloy AZ91D?
2. Provide more discussion and fundamental relations for figure 2. The meltpool “Figure 2 a” needs better quality.
3. Add some quantitative results to the abstract.
4. What is the future direction of this work?
5. Laser absorptivity in AM is important which shows the quality of the parts and transition from keyhole to conduction mode. Please read and add the following ref in this area. “The effect of absorption ratio on meltpool features in laser-based powder bed fusion of IN718”.
6. Please update the introduction with the new publications in the field. Authors are encouraged to read and add the following new papers in the field.
· High-cycle fatigue properties of curved-surface AlSi10Mg parts fabricated by powder bed fusion additive manufacturing
· Proposal of design rules for improving the accuracy of selective laser melting (SLM) manufacturing using benchmarks parts
· Fatigue life optimization for 17-4Ph steel produced by selective laser melting
Author Response
Dear Reviewer, The authors would like to thank you for your valuable comments which have helped to improve the quality of the paper. We have gone through all your comments and incorporated them into the revised text of the paper. Please find below our response to each comment.
- How authors provided the information in Table 1. Chemical composition of magnesium alloy AZ91D?
The chemical composition values in Table 1 were provided by the powder producer Dome Metals Co., LTD, Zhengzhou, China, while the chemical analysis of the powder was done by Atomic absorption spectroscopy. Information was added at the end of section 2.1.
- Provide more discussion and fundamental relations for figure 2. The meltpool “Figure 2 a” needs better quality.
In figure 3 (in previous version Figure 2) there are no meltpools; therefore, we assume that the comment is related to Figure 2 (in previous version Figure 1) where the melt pool is represented in pictures (a) and (b). In these pictures, the parameters that were measured to evaluate the quality of single track welds are shown.
Figure 2 in the paper is in original quality – direct output of microscope measurements.
More description about fundamental background was added to section 2.2 about Single track welds.
The results are discussed in section 3.1. where the continuity and quality of single track welds in the cross section are described. More relevant studies have been added to section 3.1. where the connection in the map of process parameters was observed.
- Add some quantitative results to the abstract.
The abstract was rephrased, now the abstract contains the main ideas of the paper and the main resulting values. The main values are expressed numerically i.e., the relative density of the samples higher than 99 %, and the mechanical properties of YS = 181 MPa, UTS = 305 MPa, and A5.65 = 5.2 %.
- What is the future direction of this work?
Chapter 3.9 Future outlook was added to discuss this in more detail.
- Laser absorptivity in AM is important which shows the quality of the parts and transition from keyhole to conduction mode. Please read and add the following ref in this area. “The effect of absorption ratio on meltpool features in laser-based powder bed fusion of IN718”.
The paper is referenced now in section 2.2, where the influence of laser absorptivity is mentioned.
- Please update the introduction with the new publications in the field. Authors are encouraged to read and add the following new papers in the field.
- High-cycle fatigue properties of curved-surface AlSi10Mg parts fabricated by powder bed fusion additive manufacturing
- Proposal of design rules for improving the accuracy of selective laser melting (SLM) manufacturing using benchmarks parts
- Fatigue life optimization for 17-4Ph steel produced by selective laser melting
The recommended papers were added to section 3.9.
Round 2
Reviewer 1 Report
As the author well address the issues, it deserves a publication in Applied science.